# Add-On Type Pulse Charger for Quick Charging Li-Ion Batteries

**Bongwoo Kwak [1,2], Myungbok Kim [1]**  **and Jonghoon Kim [2,\*]**

1   EV Component & Materials R&D Group, Korea Institute of Industrial Technology, Gwangju 61012, Korea;
    bwkwak11@kitech.re.kr (B.K.); boks@kitech.re.kr (M.K.)
2   Department of Electrical Engineering, Chungnam National University, Daejeon 34134, Korea
*   Correspondence: wdhgns0422@cnu.ac.kr; Tel.: +82-10-7766-5010

**Abstract:** In this paper, an add-on type pulse charger is proposed to shorten the charging time of a lithium ion battery. To evaluate the performance of the proposed pulse charge method, an add-on type pulse charger prototype is designed and implemented. Pulse charging is applied to 18650 cylindrical lithium ion battery packs with 10 series and 2 parallel structures. The proposed pulse charger is controlled by pulse duty, frequency and magnitude. Various experimental conditions are applied to optimize the charging parameters of the pulse charging technique. Battery charging data are analyzed according to the current magnitude and duty at 500 Hz and 1000 Hz and 2000 Hz frequency conditions. The proposed system is similar to the charging speed of the constant current method under new battery conditions. However, it was confirmed that as the battery performance is degraded, the charging speed due to pulse charging increases. Thus, in applications where battery charging/discharging occurs frequently, the proposed pulse charger has the advantage of fast charging in the long run over conventional constant current (CC) chargers.

**Keywords:** add-on pulse charger; quick charge; pulse charging; Li-ion battery

## 1. Introduction

Recently, owing to air pollution and global warming issues, interest in electric vehicles (EVs) and plug-in hybrid vehicles (PHEVs) has increased greatly. The batteries used in electric vehicles are mostly lithium-based secondary batteries because of their weight and mileage. Lithium-ion batteries have continued to gain market share in the secondary battery market over the last several years. This is because of their high quality density, low self-discharge and many applications [1,2]. Therefore, the demand for durability of lithium ion batteries is increasing [3,4]. In addition, improving the charging time and efficiency of the battery is an important issue in all battery powered applications.

The demand for fast charging is increasing in the case of batteries for electric vehicles [5]. The use of electric cars is increasing the need for fast chargers. Various charging technologies have been developed to improve the condition, life cycle, charging time and charging efficiency of lithium-ion batteries [2,6–10]. Among various technologies, constant current (CC) and constant voltage (CV) charging methods are widely used [1,11]. CC–CV charging technology, charges with CC below the threshold voltage. When the threshold voltage is reached, the current gradually falls, charging to a constant voltage. The magnitude of the CC and CV depends on the battery specifications. The disadvantage of this charging technique is that it reduces the charge current in the CV section to prevent battery damage owing to overvoltage. Therefore, the charging time is extended in this section [6]. In the case of CC–CV charging, the charging time is long because the charging speed is slow.

Another method is to charge with a current higher than the rated current for fast charging until the battery voltage reaches the set threshold. After the CC period, the charging current should be reduced

for battery safety. In this case, the charging speed of the battery increases, but the cost of the charging system increases. This is because a higher charging system is required for the battery capacity [12].

Pulse charging, which charges the battery by controlling the charging current, improves fast charging time and battery charging efficiency without significantly increasing costs [13–15]. However, to achieve the pulse charging effect, the duty and frequency of the pulse current must be appropriately selected [14]. In existing studies [2,15], the battery was charged in the form of a pulse from zero current to the charging current. The pulse frequency, duty and magnitude of the charging current affect the battery charging time. Depending on the frequency and duty of pulse charging, the charging time is shorter than that of the CC–CV charger method. In a previous study [2], adaptive pulse charging reduced the charging time by approximately 12.7%. In the study [15], the charge time was improved by 3.4% by applying the duty-varied voltage pulse charging method. However, these charging technologies require a dedicated charger with additional hardware and software.

In this paper, an add-on pulse type charging system is proposed, which can be connected to a conventional charging system and pulse the charging current. The proposed add-on pulse type charging system can control the amplitude, frequency and duty of the current pulse. By applying the proposed pulse charger, the existing charger can be applied CC–CV charging and the pulse charger method. Based on the results, the faster charging effect of the pulse charging method was verified as the performance of the lithium ion battery decreased.

## 2. Structure of Conventional Charging System

A conventional charging system consists of a pulse charger as shown in Figure 1a, and a CC–CV charger as shown in Figure 1b. Pulse charging is fast, but it is accompanied by the degradation of the battery. Therefore, selective charging should be available when only a quick charge is needed. Charging requires both pulse chargers and CC–CV chargers for existing systems.

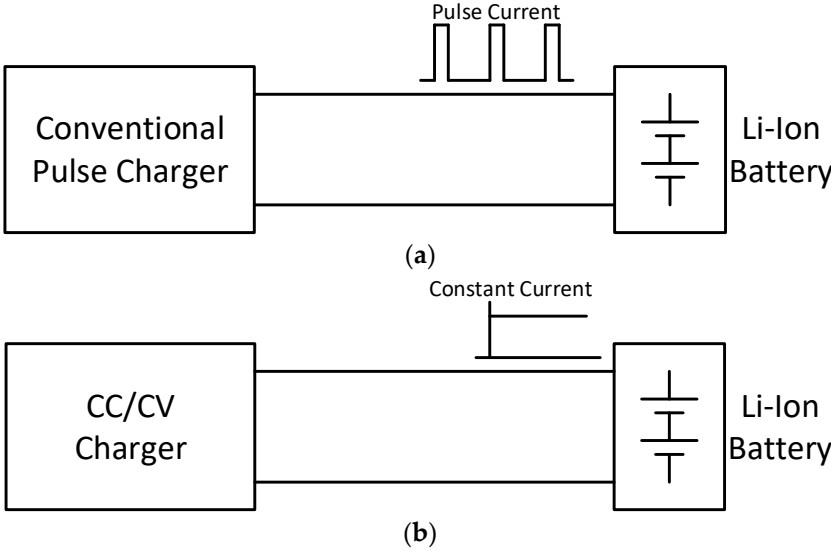

**Figure 1.** Structure of the conventional charging system. (**a**) Pulse charger (**b**) constant current–constant voltage (CC–CV) charger.

## 3. Add-On Type Pulse Charger

The proposed add-on pulsed charging system is shown in Figure 2. If the add-on type pulse charger does not operate as shown in Figure 2a, it operates as a CC–CV charger. As shown in Figure 2b, when the pulsed charging circuit is driven, it operates as a pulse charger. Therefore, the pulse charger and CC–CV charging can be selectively performed using the same charger.

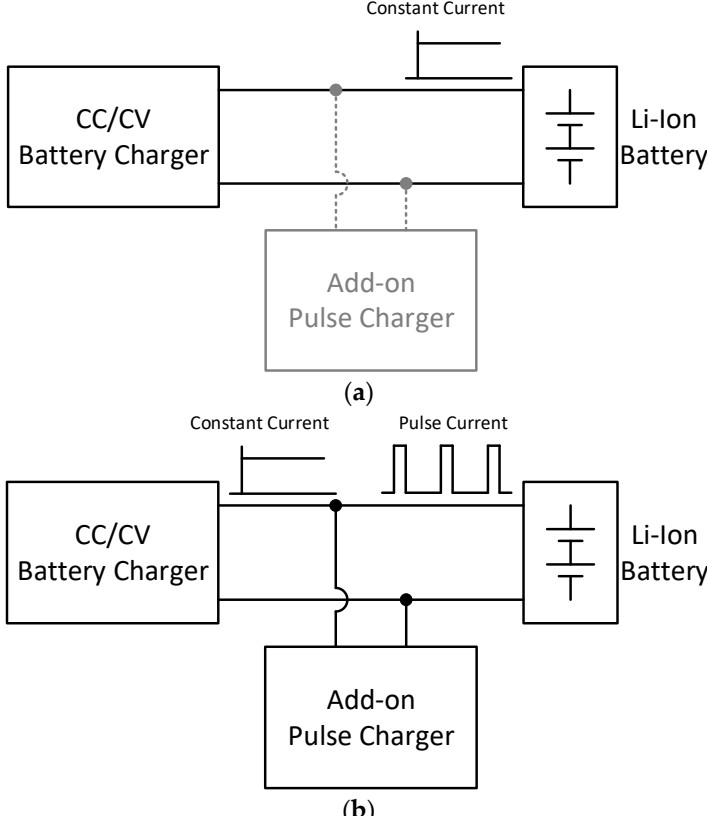

**Figure 2.** Structure of the conventional charging system. (**a**) Pulse charger (**b**) CC–CV charger.

Pulse charging uses a controlled pulse current to charge the battery. Figure 3a shows the basic structure of the pulse charger pulsed to the battery by controlling the switch ($S_1$). Figure 3b shows the waveform of the pulse current applied to the battery, which is the control element of the peak current amplitude ($\Delta I_c$), pulse duty ($D_p = t_{on}/T_p$) and frequency ($f_p = 1/T_p$). The pulse frequency, duty cycle and level affect the charging time, battery life cycle and battery parameters. As a result, pulse charging improves the charging speed and energy efficiency [5].

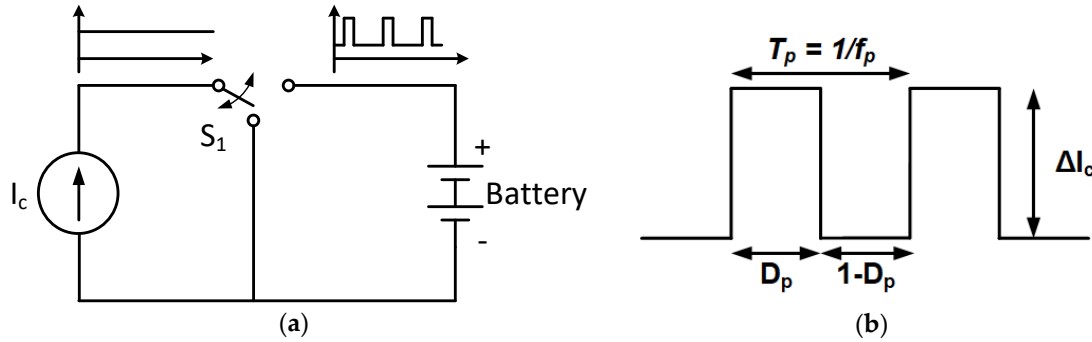

**Figure 3.** Structure of the pulse charger. (**a**) Pulse charging model and (**b**) pulse waveform.

As shown in Figure 4, a synchronous DC/DC converter circuit is applied to the pulse generator module with pulse parameter control. The basic operation uses a portion of the battery charge current to charge the link capacitor, discharging the charged capacitor energy to supply a pulsed current to the battery.

First, the link capacitor is charged using the energy of the charger. When the lower switch $Q_2$ is turned on and the upper switch $Q_1$ is turned off, it works as shown in Figure 5a. When low side switch $Q_2$ is turned off, current flows through the body diode of high side switch $Q_1$. As shown in Figure 5b,

when $Q_1$ is turned on, the voltage across $Q_1$ becomes a zero voltage condition [16]. The battery is charged by the charge current of the CC charger minus the capacitor charge current.

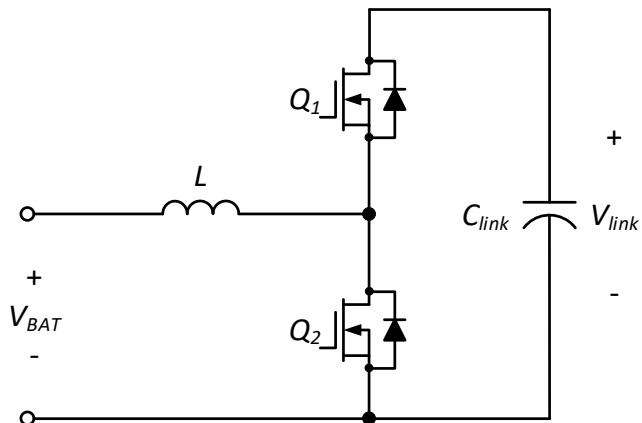

**Figure 4.** Circuit of the add-on pulse charger.

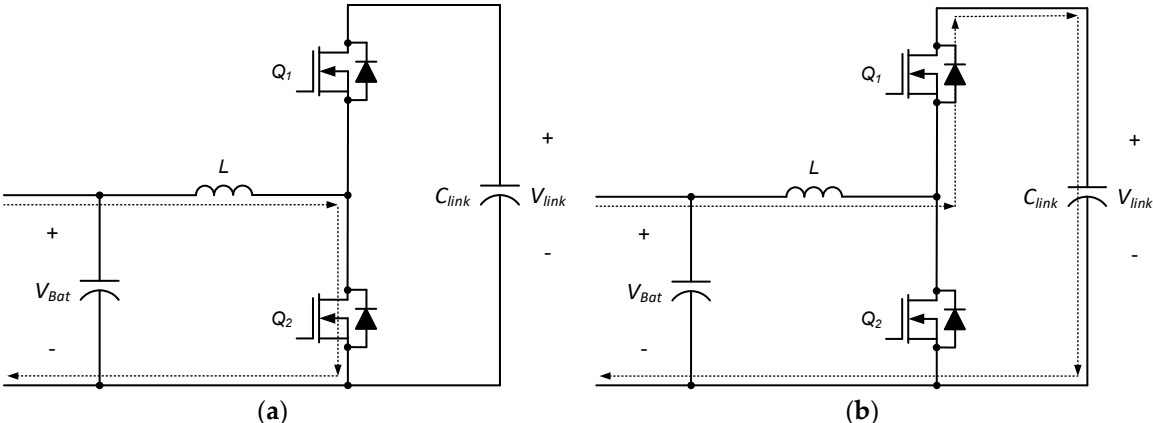

**Figure 5.** Capacitor charge mode (**a**) $Q_1$: turn off, $Q_2$: turn on, (**b**) $Q_1$: turn on and $Q_2$: turn off.

Next is the mode for discharging the link capacitor to add charging current to the battery. When the high-side switch $Q_1$ is turned on and the low-side switch $Q_2$ is turned off, it works as shown in Figure 6a. When the high-side switch $Q_1$ is turned off, current flows through the body diode of the low-side switch $Q_2$. As shown in Figure 6b, when $Q_2$ is turned on, the voltage across $Q_2$ becomes a zero voltage condition [16]. The battery is charged by the capacitor discharge current plus the charging current of the CC charger.

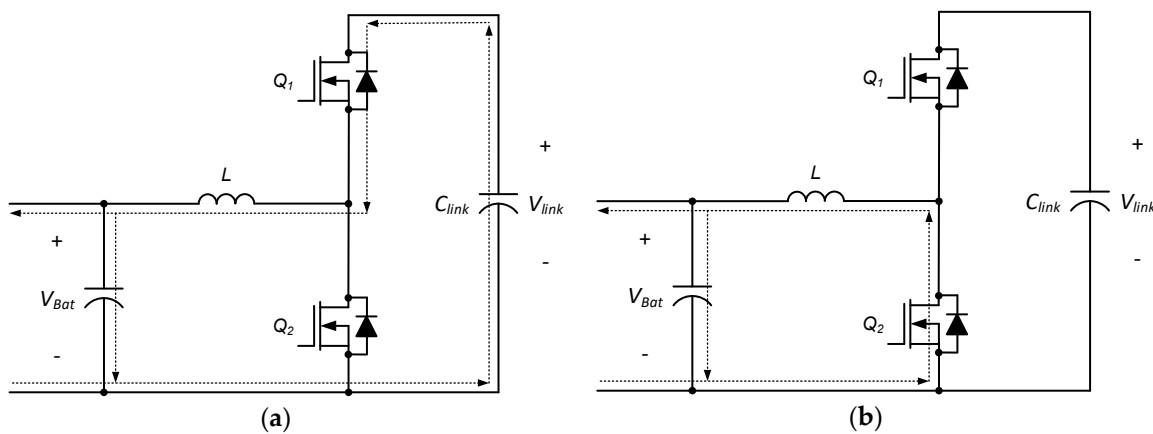

**Figure 6.** Capacitor discharge mode (**a**) $Q_1$: turn on, $Q_2$: turn off, (**b**) $Q_1$: turn off and $Q_2$: turn on.

### 3.1. Design of Circuit Parameters

Target specifications were chosen to design the pulsed charging circuit parameters. The voltage of a typical electric car is 350–400 V. Tesla organizes battery systems in series and in parallel with 18650 cylindrical cells. In this study, the battery voltage was reduced by approximately 10 times for safe experimentation. Thus, 18650 cylindrical cells were used in 10 series and two parallel configurations. The battery used was the INR 18650-35E from Samsung SDI. The battery's nominal voltage was 36 V, and the maximum charging current is 1 C-rate. The maximum voltage of the link capacitor was selected within the operating voltage range of the power semiconductor device (IPT059N15N3). Detailed specifications are listed in Table 1

**Table 1.** Specifications of the pulse charger.

| Parameter | Value | Unit |
|---|---|---|
| Battery Voltage (Nominal) | 36 | V |
| Charge/Discharge Current | 6.8 | A |
| Pulse Frequency | 200–1000 | Hz |
| Maximum Link Voltage | 120 | V |
| Switching Frequency | 100 | kHz |

The link capacitor design is important in generating the pulse current during battery charging through charging/discharging of the link capacitor. The capacitance can be calculated from the variation of the electric charge of the link capacitor. The change amount of the electric charge can be obtained through Equation (1)

$$\Delta Q = C_{link} \cdot \Delta V_{link} = \int_0^{Tp} I_L \cdot dt \tag{1}$$

where the maximum inductor current is 6.8 A. The charge/discharge pulse time is 1–5 msec. The voltage change of the capacitor through charging/discharging is shown in Equation (2).

$$\Delta V_{link} = V_{link.max} - V_{link.min} \tag{2}$$

The maximum voltage was set to 120 V, considering a 20% margin at 150 V, the rated voltage of the power semiconductor. The minimum voltage matches the battery voltage, so the voltage change on the capacitor is 84 V. Depending on the voltage specification, Equation (2) can be summarized by Equation (3).

$$V_{link.max} = V_{link.min} + \Delta V_{link} = \frac{2 \cdot I_L \cdot t_{pulse}}{C_{link}} + V_{link.min} \tag{3}$$

Equation (3) can be expressed as Equation (4) when summarized by capacitance.

$$C_{link} = \frac{2 \cdot I_L \cdot t_{pulse}}{V_{link.max} - V_{link.min}} \tag{4}$$

Through the above equation, the capacitance of the link capacitor is 160 μF. Therefore, 200 μF is chosen considering a 20% design margin.

In the charge and discharge mode, the inductor current always determines the inductance value to operate in continuous mode. For the inductor design, when the $Q_1$ turn-on time is $0 < t < DT$ in the steady state, the voltage $V_L$ applied to the inductor is given by Equation (5).

Where $D$ is the duty ratio of pulse width modulation (PWM) supplied to $Q_1$. $T$ is switching time.

$$V_L = V_{Bat} - V_{link} \tag{5}$$

The relationship between the current $I_L$ of the inductor and the inductor voltage $V_L$ is shown in Equation (6).

$$V_L = L\frac{dI_L}{dt} \tag{6}$$

Through Equations (5) and (6), the inductor current ripple for $0 < t < DT$ can be obtained as shown in Equation (7).

$$\Delta I_L = \frac{V_{link} - V_{Bat}}{L}DT \tag{7}$$

The inductor current ripple can be obtained based on the rated output current. The maximum inductor current is shown in Equation (8), and the minimum inductor current is shown in Equation (9).

$$I_{max} = I_{link} + \frac{\Delta I_L}{2} = \frac{V_{Bat}}{R} + \frac{1}{2}\left(\frac{V_{link} - V_{Bat}}{L}DT\right) \tag{8}$$

$$I_{min} = I_{link} - \frac{\Delta I_L}{2} = \frac{V_{Bat}}{R} - \frac{1}{2}\left(\frac{V_{link} - V_{Bat}}{L}DT\right) \tag{9}$$

The current ripple of the inductor can be calculated as shown in Equation (10).

$$\Delta I_L = \frac{V_{Bat}(V_{link} - V_{Bat})}{V_{link} \cdot L \cdot f_{sw}} \tag{10}$$

Equation (11) is expressed as inductance and can be obtained as follows.

$$L = \frac{V_{Bat}(V_{link} - V_{Bat})}{V_{link} \cdot \Delta I_L \cdot f_{sw}} \tag{11}$$

The inductor current ripple is typically chosen to be 10–20% of the rated current. In this study, we chose 10% to reduce the battery current ripple. Therefore, the calculated value of inductance was 370 µH. It was designed as 400 µH considering a 10% margin.

### 3.2. Control Algorithm for Add-On Pulse Charger

A control algorithm is established for charging/discharging the link capacitor for pulse current supply. If the command value of the inductor current is greater than zero, the step down operation is performed. Conversely, if the inductor current command value is less than zero, the step up operation is performed. Pulse duty and pulse amplitude commands should be set for voltage balancing of link capacitors during charge/discharge. Pulse duty is limited up to 50% and inductor current command is calculated according to pulse duty and pulse amplitude command. The current reference for the link capacitor charge mode and discharge mode is calculated by the following equation.

$$I_{L\_ref\_discharging} = I_d \times \left(1 - D_p\right) \tag{12}$$

$$I_{L\_ref\_charging} = -I_d \times D_p \tag{13}$$

where $I_d$ is the set pulse magnitude.

The inductor current, battery charge current, and link voltage waveforms for the inductor's set current are shown in Figure 7.

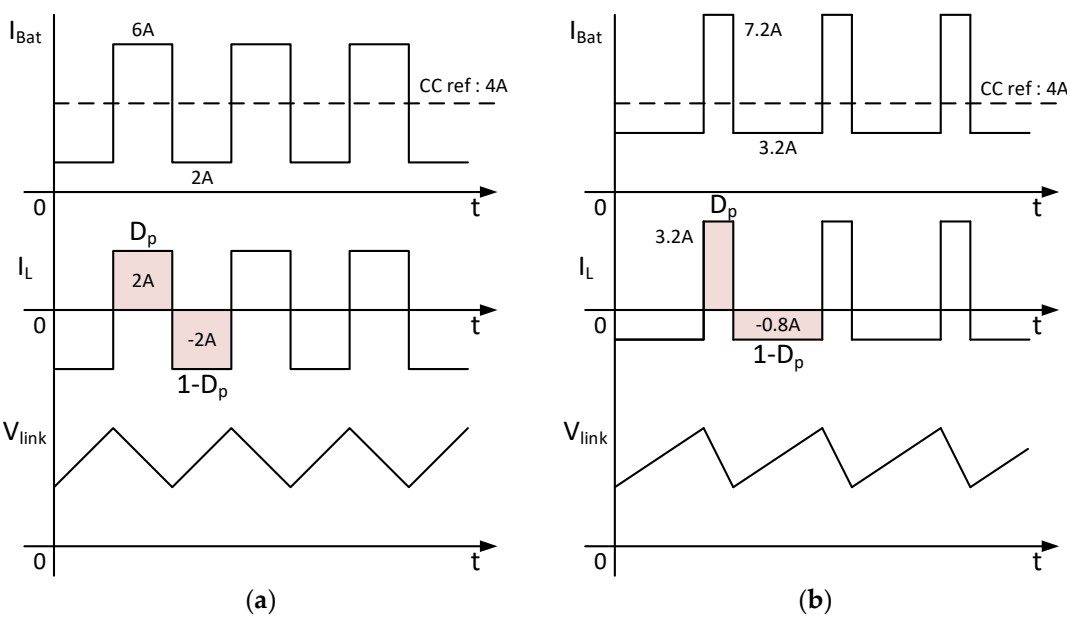

**Figure 7.** Characteristic waveforms according to the inductor current control: (**a**) $D_p$: 50% and (**b**) $D_p$: 20%.

In the 1-$D_p$ section, the link capacitor was charged using the charging current of a conventional charger. The battery was charged by subtracting the link capacitor charge current from the charger current. In the $D_p$ section, the link capacitor was discharged. The capacitor discharge current was added to the charger's charging current to charge the battery. The Proportional-Integral (PI) current controller was applied to equally control the charge and discharge current of the inductor according to the pulse duty. The control block of the pulse charger is shown in Figure 8. The inductor current reference of the pulse charger was set according to the set pulse current specification. A negative setting inductor current command charges the link capacitor. On the other hand, positive numbers discharge the link capacitor. The PI controller was used to control the inductor current for charging and discharging the link capacitor. The add-on type pulse charging system could control the pulse of the battery charging current through the inductor current control.

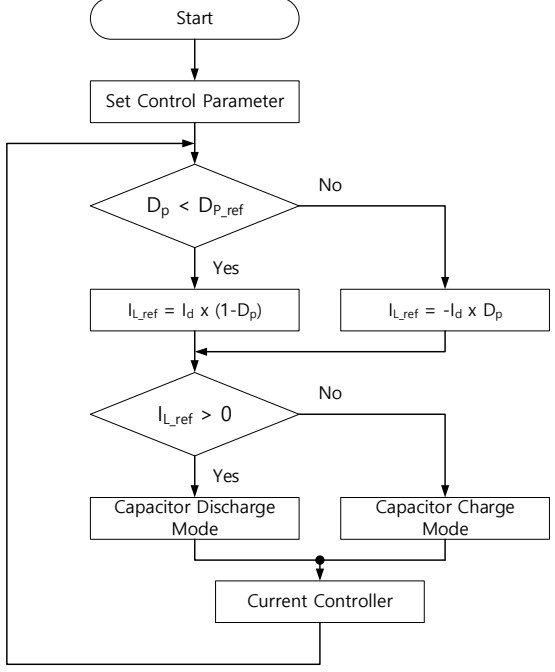

**Figure 8.** Control block diagram of the pulse charger.

## 4. Simulation and Experimental Results

This section presents the simulation and experimental results. Additional pulse charging circuits were designed using the circuit parameters selected above. The simulation used PSIM. The pulse charger specifications are listed in Table 2. The battery charge current was selected as 4 A in consideration of the INR 18650-35E maximum charge current. The simulation circuit was implemented as shown in Figure 9.

**Table 2.** Specifications and circuit parameters of the pulse charger.

| Parameter | Value | Unit |
|---|---|---|
| Inductance | 400 | μH |
| Link Capacitance | 200 | μF |
| CC Charge Current | 4 | A |
| Pulse Frequency | 200–1000 | Hz |
| Pulse Duty | 50 | % |
| Depth Pulse Current | 0–4 | A |

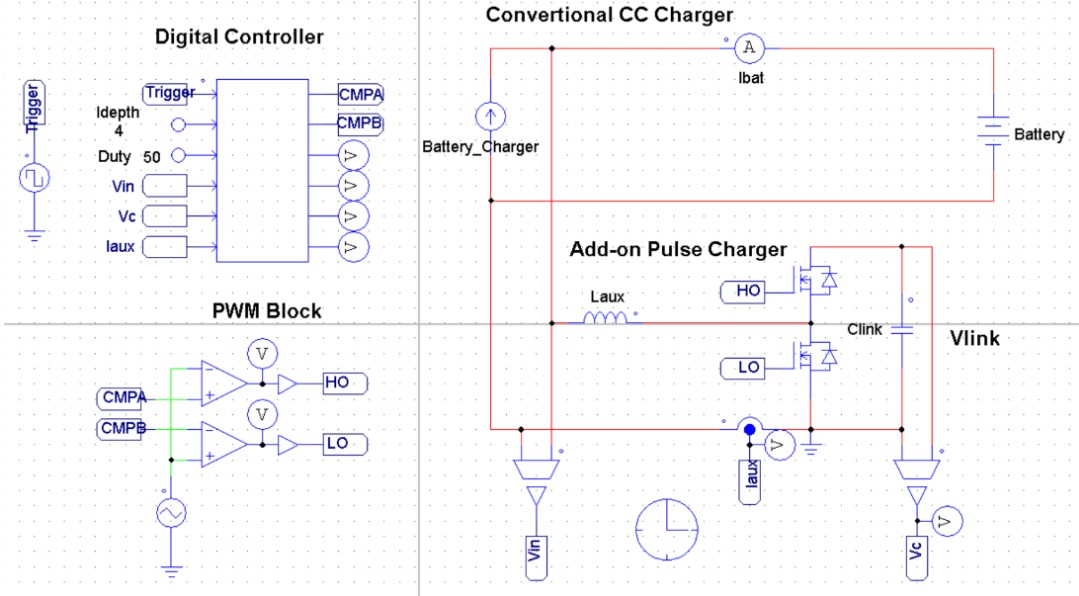

**Figure 9.** Circuit of the simulation.

The simulation results for the add-on type pulse chargers are shown in Figures 10 and 11. Figure 10 shows simulation results for a pulse frequency of 500 Hz, 50% pulse duty and an inductor current reference of 2 A. Depending on the pulse duty, the inductor current was controlled to be 1 A on charge and −1 A on discharge. Figure 10 shows the simulation results for a pulse frequency of 1000 Hz, 20% pulse duty and inductor current reference of 4 A. Depending on the pulse duty, the inductor current was controlled to be 3.2 A on charge and −0.8 A on discharge. The inverter current was controlled according to the duty and current reference of the pulse.

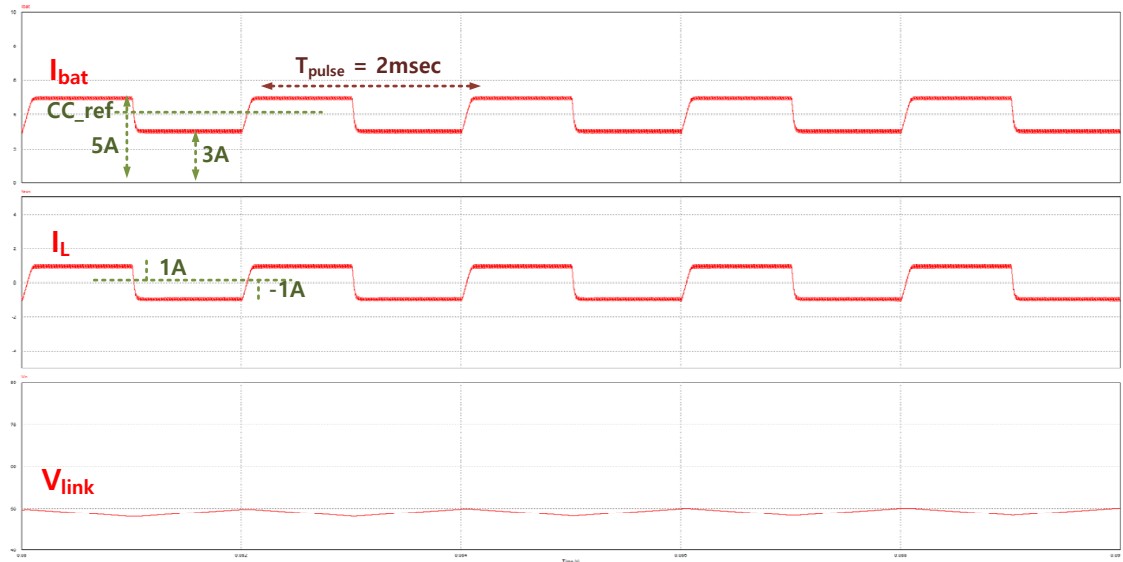

**Figure 10.** Results of the simulation at a 50% pulse duty ratio and 500 Hz pulse frequency.

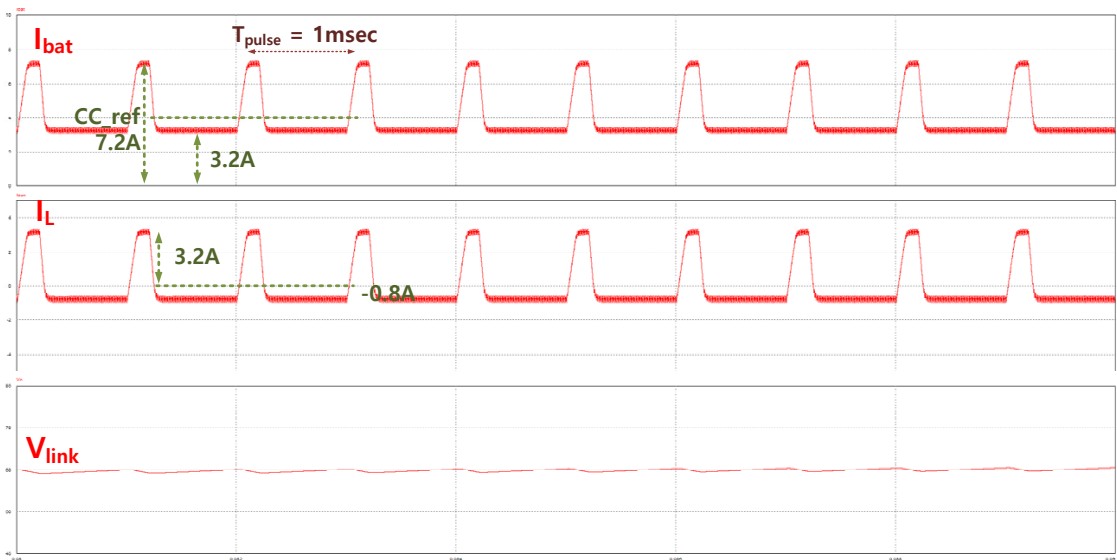

**Figure 11.** Results of the simulation at a 20% pulse duty ratio and 1000 Hz pulse frequency.

Based on the simulation, the battery was tested for pulse charging. To charge the battery with a CC, a diagram of the test equipment was constructed as shown in Figure 12a. A prototype of an add-on pulse charger is shown in Figure 12b. Figure 12c shows the configured experimental set. DC power supplies and electronic loads were used to construct a constant current charging system. The battery consisted of 10 series and two parallel structures. Oscilloscopes and data loggers were used to collect the characteristic battery data by pulse charging.

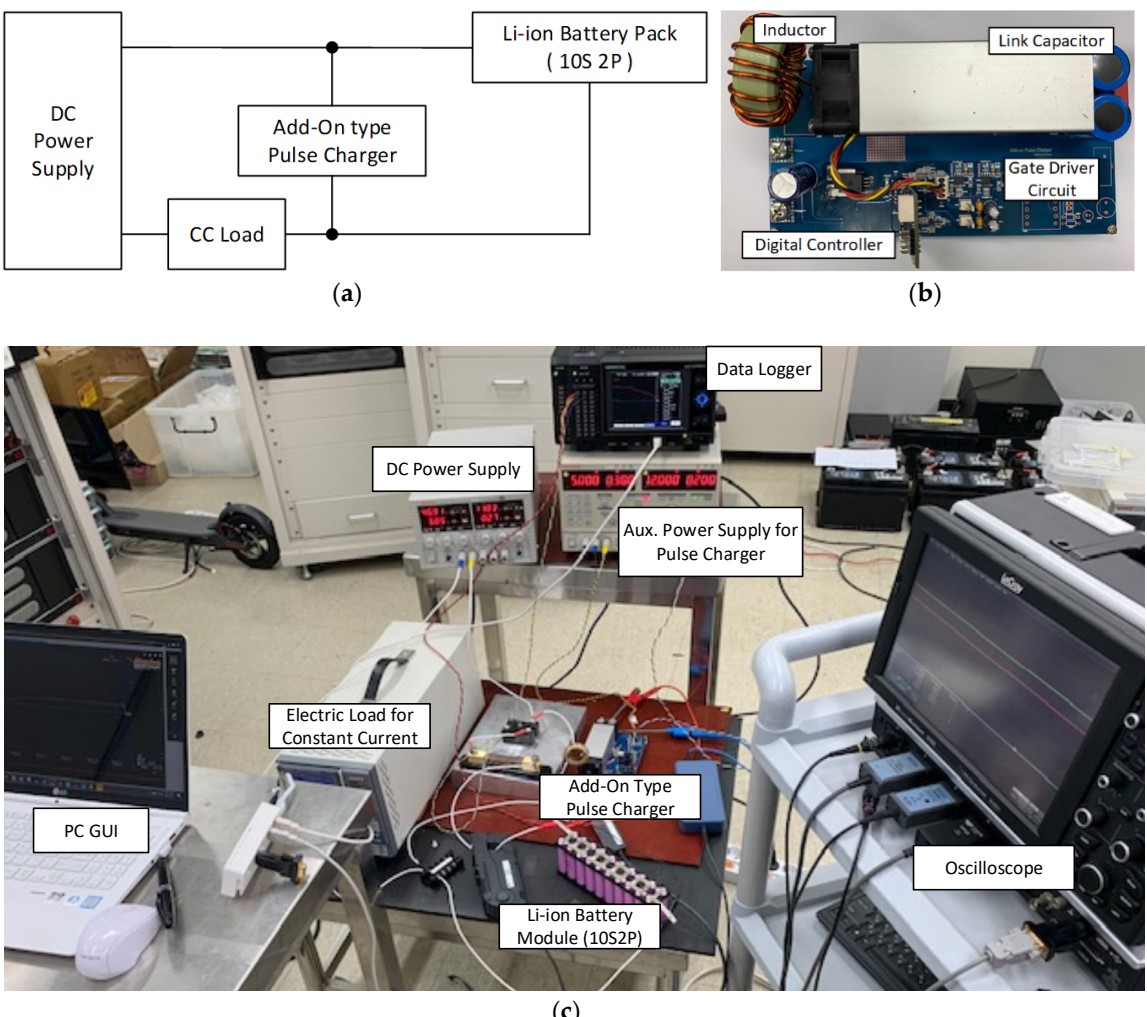

**Figure 12.** Experiment set configuration. (**a**) Block diagram of the experiment set, (**b**) prototype of the add-on type pulse charger and (**c**) experiment environment.

Pulse charge experiments were performed in the constant current charge range of the battery. CC charging and pulse charging tests were performed for a comparative analysis of conventional charging methods and pulse charging methods. The charging of the pulse was proceeded by changing the frequency, duty and magnitude of the pulse.

Figure 13 shows the steady-state characteristics by a set frequency at 50% pulse duty and 4 A pulse magnitude. Figure 14 shows the steady-state characteristics by a set duty at 1000 Hz pulse frequency and 4 A pulse magnitude. It was confirmed that the link capacitor was charged and discharged according to the set battery charge pulse. The proposed add-on pulse charger verified that the pulse frequency, magnitude and duty of the battery charge current could be selectively changed by charging/discharging the link capacitor. In addition, it could be confirmed that the mean value of the charging current of the battery is the same.

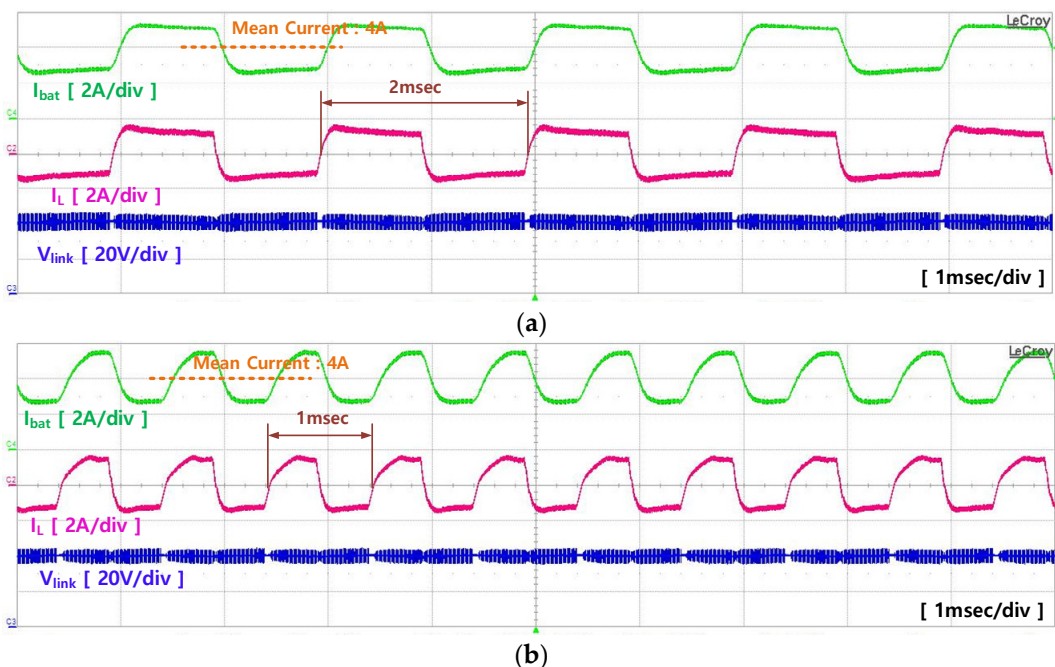

**Figure 13.** Characteristics of the steady state: (**a**) pulse frequency: 500 Hz and (**b**) pulse frequency: 1000 Hz.

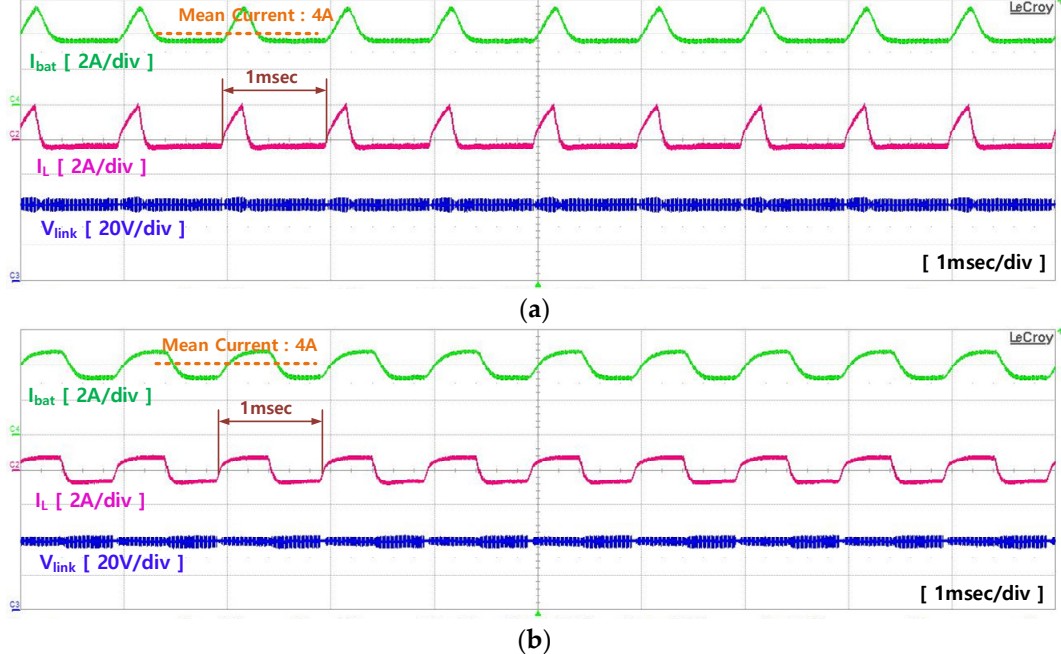

**Figure 14.** Characteristics of the steady state: (**a**) pulse duty: 20% and (**b**) pulse duty: 50%.

Table 3 shows the experimental results by the frequency change. The initial temperature is slightly different at experimental conditions. However, considering aging according to the experimental sequence, 1000 Hz was identified as the best point.

**Table 3.** Experimental results by pulse frequency.

| Pulse Frequency | Start Voltage | End Voltage | End Temperature | End Temperature | Charge Time |
|---|---|---|---|---|---|
| 500 Hz | 33.1 V | 41.5 V | 33.1 °C | 41.5 °C | 76.9 min |
| 1000 Hz | 33.1 V | 41.5 V | 30.0 °C | 40.1 °C | 77.0 min |
| 2000 Hz | 33.1 V | 41.5 V | 33.1 °C | 41.5 °C | 77.0 min |

Referring to the existing paper, it was confirmed that the optimum pulse frequency changes according to the battery characteristics. Experiments were performed to select the optimal frequency for Li-ion battery cells used in this paper. Experiments were performed in the order of 1000 Hz, 2000 Hz and 500 Hz. At this time, the pulse duty was 50% and the pulse current magnitude was 6 A. Considering the temperature conditions, all experiments were conducted in a chamber.

Next, experiments were conducted to evaluate the characteristics of the proposed pulse charging and CC charging. Pulse charging began first to reduce the effects of battery aging. The experiment proceeds in the order of pulse charging–discharging-CC charging–discharging. Pulse charge was performed at 1000 Hz pulse frequency, 50% pulse duty and 6A pulse current. Table 4 shows the pulse charging results. Table 5 shows the result of CC charging. In the above experiment, CC charging appeared faster. The reason is that due to the characteristics of the pulse controller, the average current of pulse charging was charged as low as about 0.02 A.

**Table 4.** Experimental results by pulse charging.

| No. | Start Voltage | End Voltage | Start Temperature | End Temperature | Charge Time |
|---|---|---|---|---|---|
| 1 | 33.1 V | 41.5 V | 29.6 °C | 39.7 °C | 4674 sec |
| 2 | 33.1 V | 41.5 V | 30.1 °C | 39.9 °C | 4650 sec |
| 3 | 33.1 V | 41.5 V | 30.0 °C | 40.1 °C | 4620 sec |

**Table 5.** Experimental results by CC charging

| No. | Start Voltage | End Voltage | Start Temperature | End Temperature | Charge Time |
|---|---|---|---|---|---|
| 1 | 33.1 V | 41.5 V | 30.0 °C | 39.0 °C | 4614 sec |
| 2 | 33.1 V | 41.5 V | 30.0 °C | 39.1 °C | 4584 sec |
| 3 | 33.1 V | 41.5 V | 30.0 °C | 40.2 °C | 4578 sec |

Experiments show that the CC charge was quickly charged in the new battery state. However, due to the control characteristics of the pulse charge during the experiment, the average battery charge current was 0.02 A less than the CC charge. Therefore, in the fresh battery, it was determined that the speed difference between pulse charging and CC charging was very small.

However, analysis of the experimental data confirmed that the charge rate of the pulse charge was increased when the battery was repeatedly charged/discharged. In the case of pulse charging, the second charging speed was about 24 s faster than the first charging speed. The third charge was about 30 s faster than the second charge. As the test results show above, as battery was aging, the charge speed gradually increased. In contrast, for CC charging, the second charging rate was 30 s faster than the first charging rate. The third charge was 6 s faster than the second charge. As the test results show above, as the battery was aging, the charge rate gradually slowed down. The results are shown in Table 6.

**Table 6.** Result of the reduction of the charging time by the charging method.

| Reduce the Charging Time-Pulse Charging | Reduce the Charging Time-CC Charging |
|---|---|
| 24 sec | 30 sec |
| 30 sec | 6 sec |

Pulse charging was more effective when battery performance deteriorated. In the case of a real lithium-ion battery, the fresh period is very short given the total life. In particular, in applications where charging/discharging occurs frequently, such as in electric vehicles, the proposed pulse charging method was proven to benefit from long-term fast charging rather than CC charging.

## 5. Conclusions

This paper proposed a pulse charging technique using additional charging circuits for the quick charging of lithium-ion batteries. Pulse charging shortens the charging time, but may degrade the performance of the battery. Therefore, pulse charging should be able to be optionally required. However, conventional charging systems require both pulse chargers and CC–CV chargers. The proposed add-on pulse charging circuit was connected to the existing CC–CV charger, allowing pulse charging. This charging circuit had the advantage of being applicable not only to electric vehicles but also all applications that use batteries.

This paper presented the pulse charging circuit, experimental method and data analysis. Pulse charging technology was applied to 18650 cylindrical lithium ion batteries to analyze the experimental results. Pulse charge and CC charge experiments were performed in the constant current charge section of the battery. Through repeated experiments, the charging rate by pulse charging in fresh battery condition was similar to that of conventional CC charging.

However, as battery performance deteriorated, the charge speed increase of CC charging gradually decreased. On the other hand, the increase of the charging speed by pulse charge gradually increased. To neglect the effects of aging, pulse filling was first performed. Therefore, pulse charging is more effective when battery performance degrades. In a real battery, its life is much longer than a fresh battery. Thus, in applications where charge/discharge frequently occurs, such as in electric vehicles, the proposed pulse charging rather than CC charging has a long-term advantage. In particular, the existing charging infrastructure was constructed using CC–CV chargers. Instead of installing a separate pulse charger for fast charging, using the proposed charger is expected to reduce overall costs.

In the future, we will compare pulse charging and CC charging characteristics while continuing to degrade battery performance.

**Author Contributions:** Conceptualization, B.K., and M.K.; methodology, validation, analysis, and analyzed the data, B.K., and J.K.; writing—original draft, B.K.; Writing—review and editing, supervision, and project administration, M.K., and J.K. All authors have read and agreed to the published version of the manuscript.

**Funding:** This study has been conducted with the support of the Ministry of Trade, Industry and Energy as "Future Growth Engine Business project (20003558)".

**Conflicts of Interest:** The authors declare no conflict of interest.

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
