# Peer review of "Add-On Type Pulse Charger for Quick Charging Li-Ion Batteries"

_electronics, doi:10.3390/electronics9020227_

Round 1
Reviewer 1 Report
The work investigates how to shorten LIBs charing time. The authors proposed to adopt the add-on type pulse charger to achieve the above goal. The topic is worth exploring and I enjoyed reading it. The literature review needs to be strengthened and the following related works should be cited, doi: 10.1149/07711.0257ecst, doi: 10.1149/2.040310jes, https://doi.org/10.1016/j.jpowsour.2014.06.050, doi: 10.1149/2.0521509jes, https://doi.org/10.1016/j.ijhydene.2015.04.112, doi: 10.1149/2.0291711jes, doi: 10.1149/2.1021713jes, doi: 10.1149/06127.0043ecst, https://doi.org/10.1016/j.applthermaleng.2019.114648, https://doi.org/10.1016/j.est.2019.100921, https://doi.org/10.1016/j.jpowsour.2018.06.066. In addition, there are some issues to be fixed.
does it make any economic sense by achieving a 5% increase in charging speed while potentially reducing battery Specific Energy as well as increasing total cost? When using the pulse charging, how much does the temperature rise? since it may introduce some of the battery degradations as discussed in the above-suggested papers.Author Response
Please see the attachment.

Reviewer 2 Report
Below are the technical remarks and disadvantages:
The paper conclusion is to short. Extend the conclusion to clarification of the outcomes of the research. Grammatical and typing errors should be corrected. The tags in the text align with the instructions for writing the paper. The size and unit tags are aligned with the instructions for writing the paper. The authors should highlight better the advantages of proposed approach from the point of view of economic and technical benefits in the conclusions of paper. Benefits of using numerical analysis in this type of research are not explain.Author Response
Please see the attachment.

Reviewer 3 Report
As comments to the authors, I suggest:
It seems that reference 12 is incorrect. It seems to discuss minimizing damage to the battery while cycling, not accelerating charging. BTW, if someone found a way to improve charging times by 47% without damage to the battery it would be major news...
You present the results of charging for a single cell/battery pack/experiment. In my opinion, the results then are no better than a possibility... I would encourage you to always replicate (several times if possible) the experiments so the results can be bolstered by statistical analysis.
Likewise, the description of experimental approaches should be better described. For example, how is the initial state of charge determined? Was the sequence of experiments random, or were the experiments in a specific sequence where the number of the experiment can be confounded with actual parameters in the experiment, and therefore appear as significant results?
For example, if the three experiments started from slightly different initial conditions and were carried out sequentially, that could appear to be an improvement in the charging time... You will notice that the lines in the graphs diverge significantly in the first, say, 10 minutes or so, and that the divergence then remains more-or-less constant. Careful analysis of the data may prove me wrong, and improve the quality of the results you are presenting.
Or if the battery pack was slightly damaged in a prior charging experiment and had their capacity slightly decreased, then the maximum charging voltage could be achieved faster...
For these reasons, I really encourage you to replicate and randomize the experiments.
Therefore, for the purpose of improving this paper, please at lest describe much better the approach taken to set up the experiment and collect the data, and improve the experimental results analysis instead of the minimal analysis reported.
Round 2
Reviewer 1 Report
Thank you for the update